# CoIn: A Lightweight and Effective Framework for Story Visualization and Continuation

## ABSTRACT

Story visualization aims to generate realistic and coherent images based on multi-sentence stories. However, current methods face challenges in achieving high-quality image generation while maintaining lightweight models and a fast generation speed. The main issue lies in the two existing frameworks. The independent framework prioritizes speed but sacrifices image quality with the non-collaborative image generation process and basic GAN-based learning. The autoregressive framework modifies the large pretrained text-to-image model in an auto-regressive manner with additional history modules, leading to large model size, resource-intensive requirements, and slow generation speed. To address these issues, we propose a lightweight and effective framework, namely CoIn. Specifically, we introduce a Context-aware Story Generator to predict shared context semantics for each image generator. Additionally, we propose an Intra-Story Interchange module that allows each image generator to exchange visual information with other image generators. Furthermore, we incorporate DINOv2 into the story and image discriminators to assess the story image quality more accurately. Extensive experiments show that our CoIn keeps the model size and generation speed of the independent framework, while achieving promising story image quality.

## CCS CONCEPTS

• **Computing methodologies** → *Computer vision.*

## KEYWORDS

Generative Models, Story Visualization, Story Continuation

## 1 INTRODUCTION

Story visualization is a challenging task that aims to generate coherent and visually appealing story images based on a sequence of story descriptions. The task requires the model to capture the essence of the story and the relationships between different story descriptions in the story and then translate them into realistic images that tell the story in a coherent manner. Some recent works extend the task to story continuation which provides an initial frame of the story image and generates subsequent frames based on the remaining story descriptions. Due to the practical application, story visualization and continuation have recently become an active research area [16, 18, 20, 22, 26].

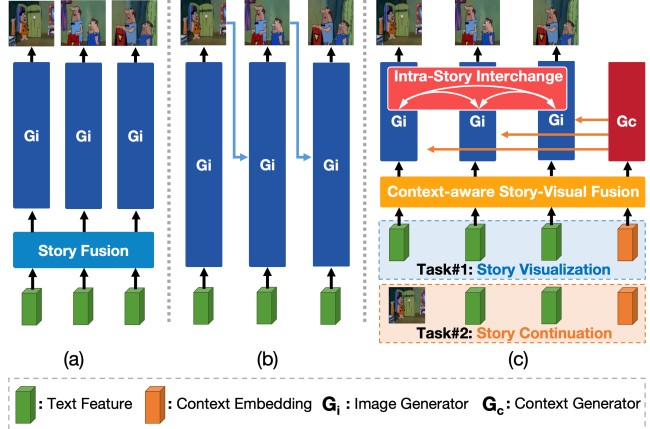

**Figure 1: (a) Existing story visualization models [16–18] always employ an independent generator design. (b) Recent models [22, 26] employ large pretrained generative models and synthesize story images frame-by-frame. (c) Our CoIn proposes a novel contextualize and interchange story generator to share the global context visual semantics and interchange local visual features during the synthesis process.**

Existing story visualization models [2, 18, 20, 21, 26, 29, 38] have significantly advanced in generating story images. However, they still face difficulties in achieving both high-quality image generation and maintaining lightweight models with a fast generation speed. The existing frameworks for story visualization exhibit inherent limitations that hinder their effectiveness and efficiency. The independent framework [16, 18, 20, 21, 38], which prioritizes generation speed, compromises image quality due to its reliance on independent generators and basic Generative Adversarial Networks (GANs). As depicted in Figure 1(a), in the independent framework, the fusion of textual information for each frame occurs only before it is passed to the image generator. The image generator operates independently during the image generation process without any information exchange, thereby increasing the difficulty of synthesizing coherent story images. Conversely, the autoregressive framework [22, 26, 29] utilizes large pretrained text-to-image models [31, 32, 34], transforming them into an auto-regressive approach for generating story images (see Figure 1(b)). It incorporates an additional history module to retain and encode the generated historical frames. However, as shown in Figure 2, this approach results in significantly larger model sizes, high resource requirements for training and inference, and the frame-by-frame design further slows the generation speed. Moreover, both independent and autoregressive frameworks do not distinguish between local story and global context information. Consequently, the image generator needs to implicitly decode the entangled local story and global context features and synthesize the corresponding visual features

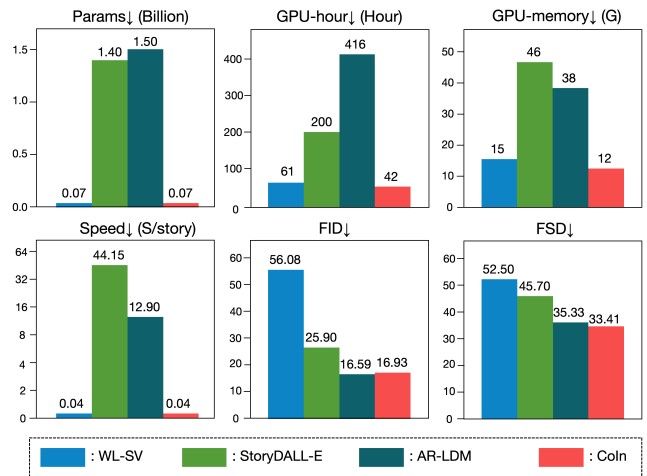

**Figure 2: Comparing with the state-of-the-art models on Pororo-SV [14]. Our CoIn achieves comparable image quality metrics, such as FID and FSD, to the large pretrained text-to-image model-based autoregressive framework (StoryDALL-E[22], AR-LDM[26]), while preserving the fast speed, small model parameters, and low resource requirements of the independent framework (WL-SV [16]).**

simultaneously. This diminishes the model's learning efficiency and compromises the quality and consistency of the generated images.

To achieve high-fidelity and coherent story image generation while maintaining reasonable computational requirements and generation speed, we propose the CoIn (Contextualize and Interchange) for story visualization and Continuation. As shown in Figure 2, compared to WL-SV[16] which is based on the independent framework, our CoIn boosts the image quality significantly while keeping the generation speed and parameters of the story generator. Compared with StoryDALL-E[22] and AR-LDM[26] which adopt pretrained text-to-image model-based autoregressive framework, our CoIn achieves competitive results with ~1000× and ~360×faster synthesis speed and 5.0% and 4.6% generator parameters.

With careful consideration of the characteristics of story visualization and continuation tasks, we develop the novel CoIn framework tailored to meet their specific requirements. Unlike previous models, our CoIn distinguishes between local story and global context to ensure the overall coherence of generated story images. As depicted in Figure 1(c), we propose the Context-aware Story Generator which consists of Contextual Story-Visual Fusion, local story image generators, and a shared story context generator. The Contextual Story-Visual Fusion fuses the text and source image features for each image generator and extracts the global context information for the context generator. Explicit extraction of global context and specialized context generator alleviate the difficulty for the Context-aware Story Generator in handling both the global context and the local story information simultaneously. Additionally, our CoIn supports information exchange between different frames of the same story during the generation process to ensure a consistent visual appearance. We propose an Intra-story Interchange module, which allows different image generators of the same story to exchange visual features, promoting visual consistency across the

generated images. Lastly, we introduce DINO-based discriminators, which incorporate the pretrained DINOv2 [25] visual backbone into the story and image discriminators to assess the quality of story images more accurately.

Overall, our contributions can be summarized as follows:

- We introduce a fast, lightweight, and effective framework for story visualization and continuation to synthesize high-quality and coherent story images.
- We propose a Context-aware Story Generator that explicitly extracts global context information and shares context features with all image generators during the synthesis process.
- We propose the Intra-story Interchange module, which enables the exchange of visual features between intra-story image generators to enhance visual consistency.
- We propose the DINO-based story and image discriminators, which assess the quality of story images more accurately.
- Extensive qualitative and quantitative experiments demonstrate that the proposed CoIn achieves promising results with fast generation speed and a much smaller model size.

## 2 RELATED WORK

**Text-to-Image Synthesis.** Recent advancements in text-to-image synthesis have primarily focused on three main frameworks: Generative Adversarial Networks (GANs) [8], autoregressive models [4, 5, 31], and diffusion models [24, 30, 32, 47]. GANs, such as Stack-GAN [45, 46], AttnGAN [42], DM-GAN [48], DF-GAN [40], GALIP [39], and StyleGAN-T [36] employ adversarial training strategies between generators and discriminators to generate high-quality images from text descriptions. Large pretrained autoregressive models, including DALL·E [31], Make-A-Scene [6], Parti [43], and VAR[41], have demonstrated scalability and proficiency in synthesizing images. These models generate images by sequentially autoregressively predicting pixel tokens based on previously generated tokens. Diffusion models [3, 12, 13, 23, 37], such as VQ-Diffusion [9], GLIDE [24], DALL-E2 [30], Latent Diffusion Models [32], Imagen [35], eDiff-I [1], and SDXL [27], have gained significant interest. These diffusion models address some of the challenges faced by GANs, such as mode collapse and training instability, resulting in the generation of diverse sets of images. These text-to-image generative models have a significant impact on story synthesis models and are often utilized as the generative backbone for generating story images based on story descriptions.

**Story Visualization and Continuation.** StoryGAN [18] was the pioneering work that introduced the task of story visualization and proposed a GAN-based sequence generation model. It consisted of a deep RNN-based context encoder and two discriminators for images and stories. Subsequent works have built upon and refined this network. For example, CP-CSV [38] introduced a new foreground segmentation module to optimize the consistency between characters and backgrounds in the story. DUCO [21] and VLC [20] [17] enhanced semantic consistency through dual learning, with DUCO considering inter-image sequence consistency via copy-transform and VLC focusing on textual information by incorporating external common-sense knowledge. WL-SV [16] and Clustering GAN [17] simplified the two-stage GAN network of StoryGAN and further improved story quality through fine-grained word-level features and

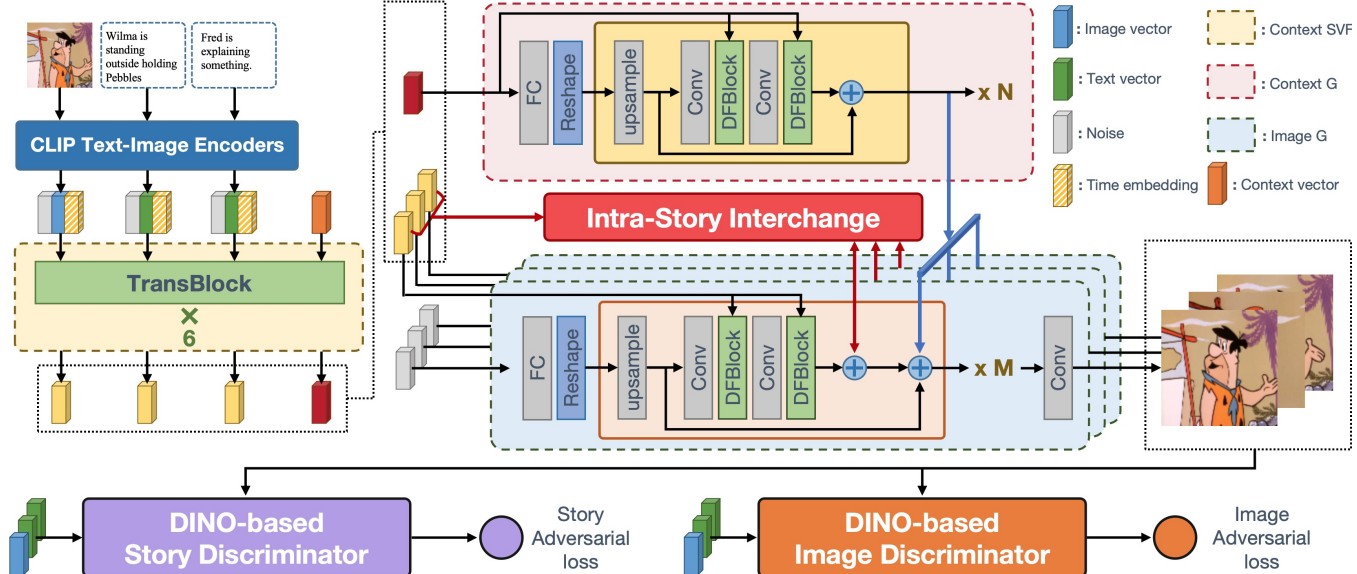

**Figure 3: The architecture of the proposed CoIn for Story Visualization and Continuation. Our CoIn proposes a novel contextualize and interchange framework to share the global context visual semantics and interchange local visual features.**

clustering learning, respectively. These GAN-based models adopt the independent framework. Conversely, recent approaches such as pre-trained DALL-E-based StoryDALL-E [22], and pre-trained stable diffusion-based AR-LDM [26] have been developed for story visualization. They utilize large pre-trained text-to-image models and synthesize story images in an autoregressive manner. Notably, StoryDALL-E [22] also introduced a new task called story continuation, which involves generating unseen plots and characters based on a given source frame. AR-LDM [26] and Make-A-Story [29] can handle both the story visualization and story continuation tasks. Recently, there are some works focusing on open-ended story visualization. TaleCrafter [7] pioneered an interactive approach with sketch and layout controls for story visualization. CogCartoon [49] introduced a character-plugin generation to minimize data and storage requirements. Intelligent Grimm [19] curated a diverse open-ended story dataset from YouTube and e-books.

The proposed CoIn differs greatly from the previous story visualization and continuation models. The CoIn shows a novel contextualize and interchange framework that is different from independent and autoregressive frameworks. It decomposes local image generation and context generation, which enables the story generator to synthesize more coherent story images. Furthermore, it adopts an Intra-Story Interchange module to exchange visual information between intra-story generators. Compared to previous models, our CoIn is a more effective and efficient framework for synthesizing high-quality and coherent story images.

## 3 THE PROPOSED METHOD

In this work, we propose a novel framework for story visualization and continuation named CoIn. To synthesize high-quality story images while maintaining reasonable computational requirements and generation speed, we propose: (i) a Context-aware Story Generator (Context-aware SG) that explicitly captures and leverages global context information by sharing contextual features with all image generators throughout the synthesis process. (ii) an Intra-story Interchange module (Intra-SI) that facilitates the exchange of visual features among image generators within the story to improve visual consistency. (iii) a pair of DINO-based story and image Discriminators (DINO-baed D) that enhance the accuracy in assessing the quality of story images. In the following section, we first present the overall structure of our CoIn. Then, we introduce Context-aware SG, Intra-SI, and DINO-baed D in detail.

### 3.1 Model Overview

As illustrated in Figure 3, CoIn consists of CLIP Text and Image Encoders [28], a Context-aware SG with a Contextual Story-Visual Fusion module (Contextual SVF), an Intra-SI module, a pair of DINO-based story and image discriminators. The CLIP Text-Image Encoder encodes text descriptions and source story images into text and image vectors. The Contextual SVF takes text and image vectors, Gaussian noise, and a learnable context vector as input and outputs the fused story embedding and extracted context embedding. Afterward, the Context-aware SG utilizes the noise, fused story embedding, and extracted context embedding to synthesize story images. The story generator comprises two classes of sub-generators: image generator and context generator. The image generator employs fused story embedding, while the context generator utilizes extracted context embedding. In the image generator, the noise is passed through a Fully Connected (FC) layer and reshaped to $(4, 4, 256)$. Then, a series of upsampling blocks upsamples the image features and incorporates fused story embedding into the generation process. The Intra-SI collects synthesized image features from different image generators within the same story and performs cross-frame visual feature exchange. The contextual visual features synthesized by the context generator are added to the output features in the upsampling block. Finally, a convolution

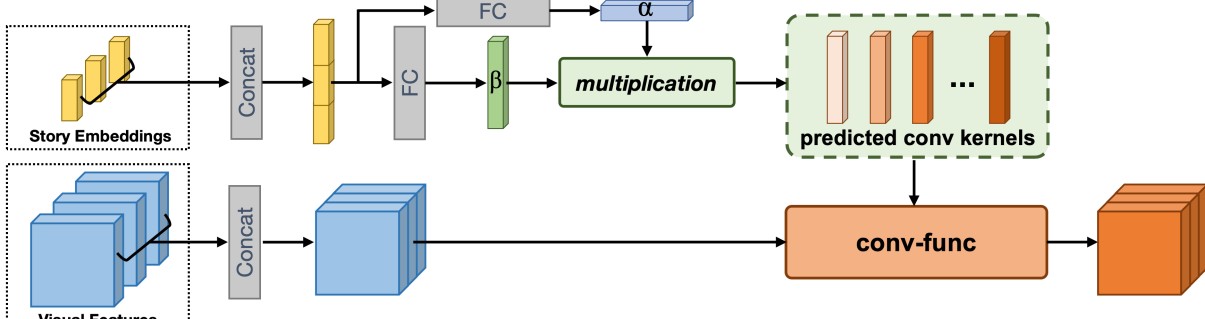

**Figure 4: Illustration of the Intra-Story Interchange module which exchanges visual features among image generators.**

layer converts the image features into RGB images. DINO-based Story Discriminator and Image Discriminator take the generated and real story images as input and extract their visual features through the frozen DINOv2 visual backbone [25]. Then, they assess the quality of story images based on the extracted visual features along with text and image vectors. By distinguishing synthesized story images from real ones, the discriminators promote the story generator to synthesize higher-quality story images.

## 3.2 Context-aware Story Generator

In this section, we detail the proposed Context-aware SG. The Context-aware SG differs from the previous independent and autoregressive generators. It shows a novel story generator that explicitly captures and leverages global context information by sharing contextual features with all image generators during the synthesis process. The Context-aware SG comprises three key components: a Contextual Story-Visual Fusion module, a Shared Context Generator, and Story Image Generators.

**Contextual Story-Visual Fusion.** The Contextual SVF aims to fuse the given text and image features and extract the context information from given conditions. However, the inputs of the story visualization and continuation are different. For story visualization, the input is a sequence of encoded text features of story descriptions, where each description describes the visual content of the corresponding frame in the story. For story continuation, the input consists of two parts: the encoded visual features of source story images and the encoded text features of story descriptions corresponding to the last frames. As shown in Figure 3, the Contextual SVF takes the pre-trained CLIP model [28] to encode source story images and story descriptions. Due to the large pre-training of image-text contrastive learning, CLIP connects the image and text spaces. The encoded text and image vectors are effectively aligned in the shared semantic space, enabling our Contextual SVF to simultaneously process the image and text features extracted by CLIP. Benefiting from this, our CoIn unifies the story visualization and continuation tasks in one model. Then, the encoded text and image vectors are concatenated with a Gaussian noise vector and combined with the time embedding. This integration allows for image variance in the synthesized story images, while the time embedding incorporates temporal information. Additionally, a learnable context vector is appended to the inputs. This vector

is specifically designed to extract context information from both source story images and descriptions.

Unlike previous models that implicitly fuse context information into the story embedding, our Contextual SVF explicitly extracts context information. This explicit extraction enables Contextual SVF to separate the global context information from the local story features. The Contextual SVF consists of six stacked transformer blocks, each containing eight attention heads. This transformer-based architecture facilitates parallel processing and ensures scalability for story lengths of varying sizes. With contextual story-visual fusion and explicit context extraction in place, the Context-aware Story Generator obtains fused story embeddings and extracted context embeddings. These embeddings are utilized by the image generators and the context generator, enabling them to generate story images that are coherent and contextually grounded.

**Story Image Generator.** The structure of the Story Image Generator is shown in Figure 3, as highlighted by the blue dashed box. The Gaussian noise vector is fed into the FC layer and reshaped to $(4, 4, 256)$ as an initial image feature. Then we apply a sequence of Story Fusion blocks (SF-BLKs) to upsample the image feature to the target image size and fuse story and context information into synthesized story images. The SF-BLK is composed of one upsample layer, two convolution layers (Conv), and two deep text-image fusion blocks (DFBlock) [40]. As shown in recent text-to-image works [40], DFBlock can fuse text and image features effectively through deep affine transformations. Thus, we employ it to fuse story features and intermediate visual features. After the fusion of the story and visual features, the SF-BLK sends the fused visual features to the Intra-SI and receives the exchanged visual features from it. The exchanged visual features are added to the fused visual features. Lastly, the SF-BLK combines the context visual features generated by the Context Generator and the original features from the shortcut before fusion. Through SF-BLK, the output visual features contain local story information, global context information, and cross-frame visual information. There are $M$ SF-BLKs stacked in the Image Generator, and one convolution layer converts the fused image features into RGB images.

**Shared Context Generator.** The structure of the Shared Context Generator is also shown in Figure 3, as highlighted by the pink dashed box. To ensure the generation of story images with consistent overall context, the Context Generator and Image Generator are designed with a similar structure, facilitating the effective fusion

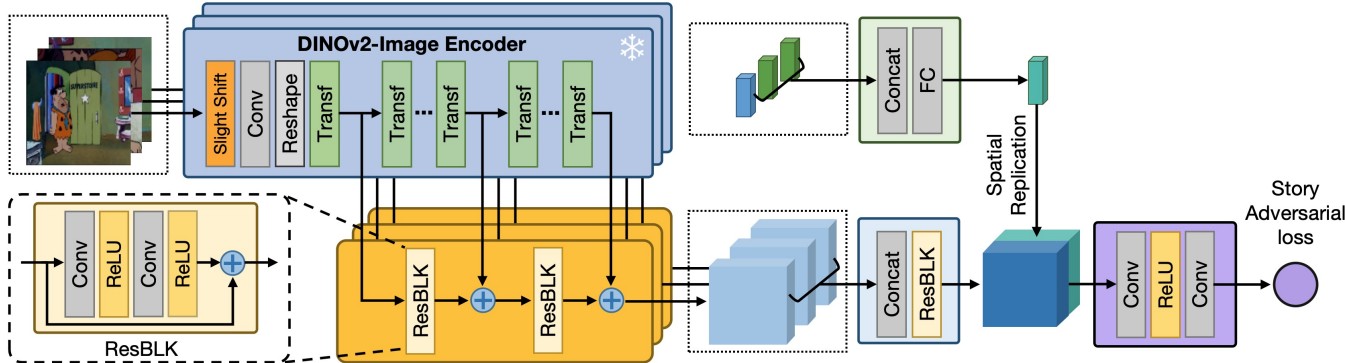

**Figure 5: Illustration of the proposed DINO-based Story Discriminator for story visualization and continuation tasks.**

of context information. However, unlike Image Generator, Context Generator only takes the context embedding as input. There are $N$ Context Fusion blocks (CF-BLKs) stacked in the Context Generator. As SF-BLKs, we adopt the DFBlock[40] to fuse the extracted context embedding at each CF-BLK. Then, the context visual features generated by the Context Generator are shared with all Image Generators. The Image Generator and Context Generator do not share parameters, allowing for a more thorough learning of image and context generation. To strike a balance between sharing story context and preserving different visual details in each story image, the context visual features are only introduced in the first 3 SF-BLKs. Compared to previous methods, the Context Generator enhances the global context information and alleviates the difficulty of the story generator in synthesizing coherent story images.

### 3.3 Intra-Story Interchange

To enable feature interchange during the intra-story image generation process, we propose the Intra-SI module. As illustrated in Figure 3 and Figure 4, the Intra-SI takes the fused story embeddings from the Contextual SVF and collects the image visual features from intra-story image generators. First, Intra-SI concatenates the story embeddings and visual features, respectively. The concatenated story embeddings are then fed into two FC layers to predict the vector $\alpha \in \mathbb{R}^C$ and $\beta \in \mathbb{R}^C$, where $C$ is the channel size of the concatenated visual features. Afterward, the vectors $\alpha$ and $\beta$ are reshaped to $(C, 1)$ and $(1, C)$. We perform matrix multiplication between the reshaped $\alpha$ and $\beta$ to obtain the predicted $1 \times 1$ convolutional kernel weights. Finally, we conduct the $1 \times 1$ convolution process on concatenated visual features based on the predicted kernel weights. As part of the Shared Context Generator, the Intra-SI modules are applied only in the first four SF-BLKs to strike a balance between visual appearance consistency and variance. The proposed Intra-SI module in our CoIn framework facilitates interaction among image generators throughout the story generation process. By incorporating the Intra-SI module, the image generators have the opportunity to exchange visual features among themselves. This exchange of visual features enhances the generated story images, resulting in improved visual appearance consistency.

The proposed Intra-SI enables our CoIn to interact across the image generators during the story generation process. As the Shared Context Generator, the Intra-SI modules are only applied in the first

4 SF-BLKs to balance visual appearance consistency and variance. The Intra-SI provides the chance for the image generator to exchange visual features among them and improves generated story images with better visual appearance consistency.

### 3.4 DINO-based Discriminators

To improve the efficiency of adversarial learning and the accuracy of story image quality assessment, we propose the DINO-based Discriminator. We first introduce the DINO-based Story Discriminator. As shown in Figure 5, the story image sequence is first encoded by the frozen DINOv2-Small Image Encoder. The visual features from the $2^{nd}$, $7^{th}$, $9^{th}$ layers in DINOv2-Small are extracted and further analyzed through two residual blocks. Then we concatenate the output visual features and encode them into a more compact visual feature. We also concatenate the text and image vectors and encode them into a more compact vector. Afterward, we replicate the encoded text-image vector and concatenate it with compact visual features. A story adversarial loss is predicted by two convolution layers to evaluate the story quality. Based on the visual understanding ability of pre-trained DINOv2 [25], the DINO-based Story Discriminator can assess the story quality more accurately. The DINO-based Image Discriminator shares a similar structure but differs in how it processes the input features. Rather than concatenating the visual features of the same story as the whole story visual feature, it concatenates the visual features of each image with their corresponding textual features. This allows the discriminator to assess the quality of each image.

The DINO-based Discriminator is partly inspired by GALIP [39] which employs the pretrained CLIP-ViT [28] to extract visual features in the image discriminator. In this work, we extend it and propose the DINO-based Story and Image Discriminator to assess the story quality. Compared with CLIP-ViT which only utilizes high-level contrastive pretraining, DINOv2 incorporates an additional mask to reconstruct masked parts. It enables DINOv2 to extract richer and more detailed visual features. However, we observed that directly using the DINOv2 model resulted in the generation of patch-like patterns in the synthesized images, as shown in Figure 8(a). These patch-like patterns were consistent with the size of the DINOv2 input (16, 16). When we replaced DINOv2 with CLIP-ViT-B/16, the patch-like patterns disappeared. We suspect that this issue may be related to the masked pretraining employed by DINOv2,

which might not provide sufficient supervision for smooth transitions between adjacent image patches. Consequently, the generator may have learned to generate adversarial features. To address this issue, we propose the Slight Shift trick. Before feeding the images into DINOv2, we slightly enlarge them and randomly crop an image corresponding to the resolution of DINOv2's input. This process helps prevent the generator from accurately identifying the patch intervals, thereby avoiding the generation of patch-like patterns.

## 3.5 Objective Function

We follow previous text-to-image models [39, 40] and employ the hinge loss [44] and MA-GP [39, 40] to stabilize the adversarial training process. Finally, the whole formulation of our CoIn is shown as follows:

$$
\begin{aligned}
L_{D_I} = &-\mathbb{E}[min(0, -1 + D_I(x, e))] \\
&- (1/2)\mathbb{E}[min(0, -1 - D_I(G(z, e), e))] \\
&- (1/2)\mathbb{E}[min(0, -1 - D_I(x, \hat{e}))] \\
&+ k\mathbb{E}[(\|\nabla_{c_i} D_I(c_i, e)\| + \|\nabla_e D_I(c_i, e)\|)^p], \\
L_{D_S} = &-\mathbb{E}[min(0, -1 + D_S(x, e))] \\
&- (1/2)\mathbb{E}[min(0, -1 - D_S(G(z, e), e))] \\
&- (1/2)\mathbb{E}[min(0, -1 - D_S(x, \hat{e}))] \\
&+ k\mathbb{E}[(\|\nabla_{c_s} D_S(c_s, e)\| + \|\nabla_e D_S(c_s, e)\|)^p], \\
L_D = &\lambda L_{D_I} + \gamma L_{D_S}, \\
L_G = &-\lambda \mathbb{E}[D_I(G(z, e), e)] - \gamma \mathbb{E}[D_S(G(z, e), e)],
\end{aligned}
\tag{1}
$$

where $z$ is the Gaussian noise vector; $e$ is the encoded text and image vectors by the CLIP; $\hat{e}$ is the mismatch text and image vectors; $x$ is ground truth story images; $G$ is the Context-aware Story Generator; $D_I$ and $D_S$ are the DINO-baed Image Discriminator and Story Discriminator; $c_i$ and $c_s$ are two extracted visual features by the frozen DINOv2-Small in Image and Story discriminators; $k$ and $p$ are two hyper-parameters of Matching-Aware Gradient Penalty [40]; $\lambda$ and $\gamma$ are two hyper-parameters of image adversarial loss and story adversarial loss.

## 4 EXPERIMENTS

### 4.1 Datasets

We evaluate our approach on two challenging datasets: Pororo-SV [14] and Flintstones-SV [10]. The Pororo-SV dataset consists of 10191 training samples, 2334 validation samples, and 2208 testing samples. The Flintstones-SV dataset consists of 20132 training samples, 2071 validation samples, and 2309 testing samples. Each story in these datasets consists of five consecutive frames, and each image corresponds to a story description. The partition of datasets into training, validation, and testing subsets follows established practices from previous studies [16, 22, 26, 29].

### 4.2 Training and Evaluation Details

Our method is developed using PyTorch. The resolution of the output images on Pororo-SV and Flintstones-SV is $256 \times 256$. The network is trained 200 epochs and 150 epochs on Pororo-SV and Flintstones-SV. We use the Adam optimizer [15] with $\beta_1$=0.0 and $\beta_2$=0.9 to train our model. We set the learning rate 0.0001 for the generator and 0.0004 for the discriminator. The hyper-parameters

**Table 1: The comparison results of story visualization on the test set of Pororo-SV and Flintstones-SV.**

| Method | Params | Speed | Pororo-SV | | Flintstones-SV | |
|---|---|---|---|---|---|---|
| | | | FID ↓ | FSD ↓ | FID ↓ | FSD ↓ |
| StoryGAN[18] | - | - | 78.64 | 94.53 | 90.55 | 122.71 |
| CP-CSV [38] | - | - | 67.76 | 71.51 | - | - |
| VLC [20] | - | - | 94.30 | 122.07 | - | - |
| WL-SV [16] | 0.07B | 0.04s | 56.08 | 52.50 | 72.37 | 91.30 |
| Make-A-Story[29] | 1.40B | 14.00s | 27.33 | 51.20 | 36.55 | 53.10 |
| AR-LDM [26] | 1.50B | 14.50s | 16.59 | 35.33 | 23.59 | 39.70 |
| CoIn (Ours) | 0.07B | 0.04s | 16.93 | 33.41 | 24.52 | 36.15 |

**Table 2: The comparison results of story continuation on the test set of Pororo-SV and Flintstones-SV.**

| Method | Params | Speed | Pororo-SV | | Flintstones-SV | |
|---|---|---|---|---|---|---|
| | | | FID↓ | FSD↓ | FID↓ | FSD↓ |
| StoryDALL-E [22] | 1.30B | 44.15s | 25.90 | 45.70 | 26.49 | 54.30 |
| MEGAStoryDALL-E [22] | 2.80B | 87.40s | 23.48 | - | 23.58 | - |
| Make-A-Story[29] | 1.40B | 12.15s | 22.66 | 44.22 | 23.74 | 52.08 |
| AR-LDM [26] | 1.50B | 12.90s | 17.40 | 37.52 | 19.28 | 43.32 |
| CoIn (Ours) | 0.07B | 0.04s | 18.63 | 34.73 | 19.95 | 39.17 |

of the generator $M$ and $N$ are set to 6 and 3. The hyper-parameters of the objective function $k$, $p$, $\lambda$, $\gamma$, are set to 2, 6, 0.5, and 0.5. All models were trained on $8 \times$ A6000 GPUs. Following the previous story visualization works [16, 22, 26], we adopt the Fréchet Inception Distance (FID) [11] to evaluate the image quality of synthesized story images, and adopt the Fréchet Story Distance (FSD) [16, 38] to evaluate the consistency of the story image sequence. To compare the efficiency of different models, we measure the generation speed and the number of parameters in the story generator. The generation speed is evaluated on a single A6000 GPU.

### 4.3 Quantitative Evaluation

To evaluate the performance of our proposed CoIn, we compare it with several state-of-the-art story visualization and continuation methods [16, 20, 22, 26, 38] in Table 1 and Table 2. From Table 1 and Table 2, we can observe that our CoIn achieves promising FID and FSD with fast generation speed and a small number of parameters. Compared with WL-SV [16] which is the best model based on the independent framework, our CoIn significantly improves the FID and FSD of Pororo-SV and Flintstones-SV. Compared with StoryDALL-E [22], which utilizes large pre-trained DALL-E models and synthesizes story images in an autoregressive manner, our CoIn framework achieves superior results despite having significantly smaller generator parameters and faster synthesis speed. This demonstrates the effectiveness and efficiency of our approach in generating high-quality story images. Compared with AR-LDM [26] which is an autoregressive model based on the powerful stable diffusion [33], our CoIn has a significantly smaller number of generator parameters but still achieves a competitive performance.

### 4.4 Qualitative Evaluation

Figures 6 and 7 show examples of visual comparisons between our CoIn and state-of-the-art story visualization and continuation

**1.** eddy is wearing an equipment with a flashlight and two robot arms on his head . eddy 's robot hand is holding a huge lollipop .
**2.** pororo is on a sleigh being surprised looking at a huge lollipop .
**3.** pororo is on a sleigh being sarcastic looking at a huge lollipop .
**4.** pororo is on a sleigh. eddy is wearing an equipment with a flashlight and two robot arms on his head .
**5.** pororo is on a sleigh and he looks at the front part of the sleigh . eddy is wearing an equipment with a flashlight and two robot arms on his head .

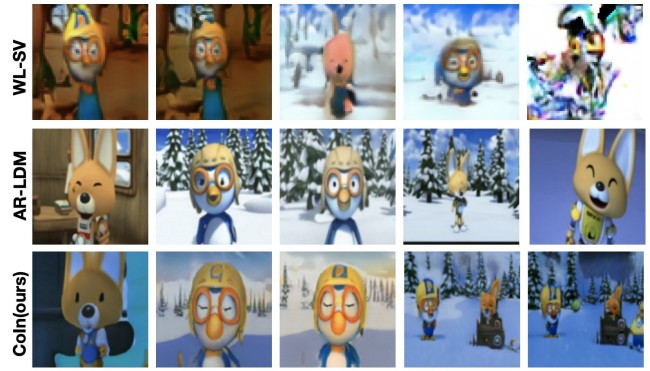

**1.** eddy is explaining to his friends that eddy was playing ball . eddy is showing the ball to his friends .
**2.** eddy is smiling and explaining that eddy was playing ball to his friends .
**3.** pororo and crong are excited to hear something from eddy .
**4.** pororo and crong are excited to hear something . pororo and crong are eager to do it themselves .
**5.** pororo and crong are very excited at an idea .

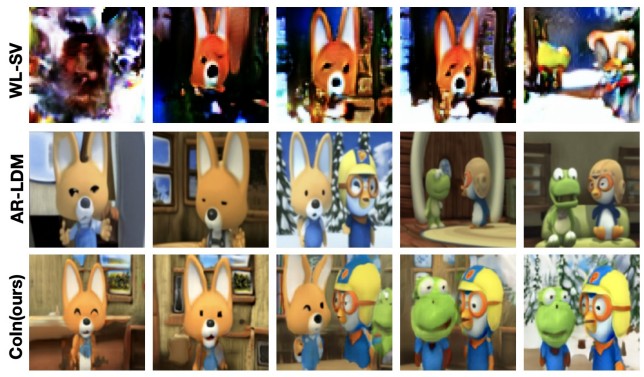

**1.** barney nods and speaks to fred , who looks angry , while standing in the living room .
**2.** fred shouts angrily in the living room
**3.** fred is in a room and is angrily talking to someone off camera left .
**4.** fred and wilma are sitting in a room . fred moves his hand out with an angry expression on his face . wilma looks at fred then turns her head back and starts talking .
**5.** fred in standing in the living room , talking to someone off screen left .

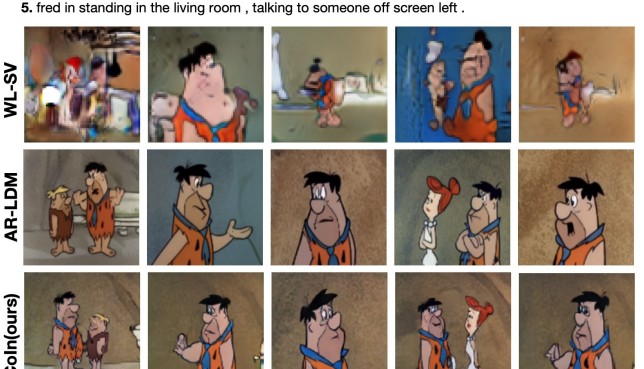

**1.** wilma and betty are sitting on the couch in the living room . betty is talking to wilma .
**2.** wilma is sitting in a room speaking to someone .
**3.** wilma is in a room talking while nodding her head .
**4.** wilma and betty are sitting on a couch in the living room . while wilma is speaking , she points her finger at betty .
**5.** wilma and betty are sitting on the couch in the living room . betty is talking and wilma is listening .

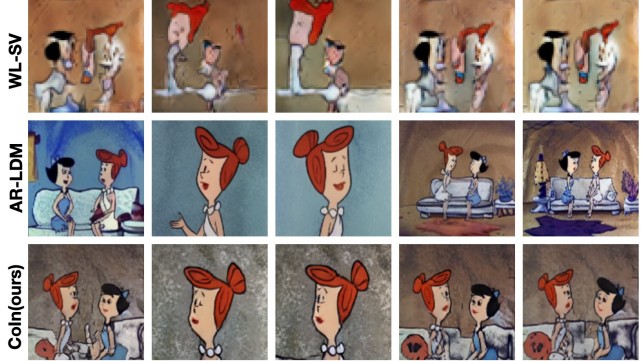

**Figure 6: Comparison of story visualization results between WL-SV, AR-LDM, and our proposed CoIn.**

models, respectively. As shown in Figure 6, the characters synthesized by WL-SV [16] contain broken or incorrect shapes. However, the story images synthesized by our CoIn have the correct shapes and clear backgrounds. Compared with StoryDALL-E [22] in Figure 7, our CoIn can synthesize a more consistent character appearance. Although our CoIn has achieved better image quality than StoryDALL-E, we have observed some limitations in terms of generalization compared to AR-LDM [26]. Due to the powerful Stable Diffusion with large-scale pretraining, AR-LDM can generate uncommon content not frequently seen in the training set. However, it is worth noting that our CoIn still achieves comparable results to AR-LDM in certain scenarios with ~360×faster synthesis speed and 4.6% generator parameters.

## 4.5 Ablation Study

To verify the effectiveness of different components in the proposed CoIn, we conduct ablation studies on the story visualization and continuation tasks, respectively. The results of Pororo-SV and Flintstones-SV are shown in Table 3. The components being evaluated include DINO-based D, Context G, Contextual SVF, and

**Table 3: The performance of different components of our model on the test set of Pororo-SV and Flintstones-SV.**

| Task | Method | Pororo | | Flintstones | |
|---|---|---|---|---|---|
| | | FID (↓) | FSD (↓) | FID (↓) | FSD (↓) |
| Visualization | Baseline | 52.64 | 65.08 | 58.29 | 64.78 |
| | + CLIP-based D | 31.46 | 52.19 | 38.71 | 55.46 |
| | + DINO-based D | 28.46 | 49.34 | 33.71 | 52.46 |
| | + Context G | 24.39 | 42.56 | 29.50 | 46.55 |
| | + Contextual SVF | 19.38 | 38.19 | 27.96 | 39.32 |
| | + Intra-SI (CoIn) | **16.93** | **33.41** | **24.52** | **36.15** |
| Continuation | Baseline | 57.15 | 66.12 | 62.11 | 67.18 |
| | + CLIP-based D | 36.18 | 52.10 | 40.66 | 58.69 |
| | + DINO-based D | 32.74 | 48.49 | 35.23 | 53.69 |
| | + Context G | 26.83 | 43.41 | 28.97 | 48.40 |
| | + Contextual SVF | 23.53 | 38.14 | 23.22 | 45.69 |
| | + Intra-SI (CoIn) | **18.63** | **34.73** | **19.95** | **39.17** |

Intra-SI. We also compare the results of CLIP-based D and DINO-based D. Our baseline is a modified text-to-image DF-GAN [40] adapted for story visualization tasks. The baseline fuses the story information through FC layers.

**1.** pororo talks about what pororo did to crong .
**2.** pororo and pororo friends are gathered at playground .
**3.** pororo and pororo friends are talking about loopy .
**4.** pororo and pororo friends are talking about pororo being secret friend .
**5.** pororo and pororo friends are talking about pororo being secret friend .

**1.** eddy is explaining to rody how to play ball . rody and eddy are both waring a glove .
**2.** rody is holding glove . rody is trying to understand how to play ball .
**3.** eddy and rody decide to play ball together . eddy and rody look excited .
**4.** eddy is holding the ball in his hand . eddy is waring a glove . rody is on the other side . eddy is about to throw the ball .
**5.** rody is making gesture of catching the ball . rody is getting ready for catching the ball with the glove .

**1.** wilma is speaking to someone in the room .
**2.** wilma is in her room . she is happy about something
**3.** wilma is standing in a room . she is talking
**4.** wilma is in a room . fred seems mad at her
**5.** fred is looking angrily at wilma while pointing a finger as they stand in a room .

**1.** wilma and betty are in the living room . wilma is standing behind a brown box .
**2.** betty is sitting on the floor in a room . she is talking and swings a paddle with writing on it .
**3.** betty is talking and going through a closet .
**4.** wilma and betty are in the living room . betty speaks . then wilma reaches down into a box .
**5.** wilma pulls a jersey out of a box while talking to betty in the living room .

**Figure 7: Comparison of story continuation results between StoryDALL-E, AR-LDM, and our proposed CoIn.**

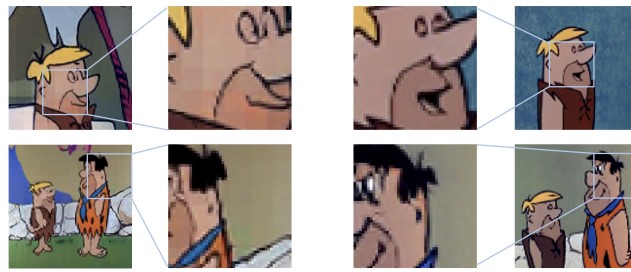

**(a) Without Slight Shift in DINO-based D**       **(b) With Slight Shift in DINO-based D**

**Figure 8: Comparison of the synthesized images between the DINO-based D with and without Slight Shift.**

From Table 3, we can observe that our proposed DINO-based D achieves better performance than CLIP-based D on these two tasks. If we further introduce the Context G to decompose the generation task, we can observe a further improvement of FID and FSD. Armed with Contextual SVF, the model also decreases FID and FSD from 24.39 and 42.56 to 19.38 and 38.19 for the story visualization task on Pororo. The proposed Intra-SI further decreases FID and FSD

on two datasets. The ablation studies demonstrate the effectiveness of our proposed modules in both story visualization and continuation tasks. Furthermore, we compare the differences in generated images between the DINO-based D with and without Slight Shift in Figure 8. It is evident that Slight Shift effectively alleviates the issue of generating patch-like patterns.

## 5 CONCLUSION

In this paper, we propose a lightweight and effective CoIn (Contextualize and Interchange) framework for story visualization and continuation. Moreover, we propose a Context-aware Story Generator that explicitly extracts the global context information and synthesizes context visual features for each image generator. Furthermore, we propose an Intra-story Interchange module to enable information exchange between intra-story image generators. Lastly, a pair of DINO-based discriminators is introduced to assess the story quality more accurately. Our CoIn achieves competitive synthesis quality with ~360×faster synthesis speed and 4.6% generator parameters. Our CoIn also significantly reduces the required resources and training time. This is particularly crucial for tasks like story visualization that involve generating a large number of images.

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
