# OpenReview forum: "CoIn: A Lightweight and Effective Framework for Story Visualization and Continuation"
_acmmm.org/ACMMM/2024/Conference — MM2024 Poster_

### Official Review · Reviewer_xhrW · 2024-05-24

**Rating:** 3
**Confidence:** 3

**Summary:**

The article introduces a lightweight and effective framework, CoIn, for Story Visualization and Continuation. This framework utilizes a GAN-based approach, extracts global context information through a Context-aware Story Generator, and visual information through an Intra-story Interchange module. It also employs DINO-based discriminators to enhance GAN training. As the article demonstrates, this method can achieve impressive visual results while significantly accelerating compared to methods with similar performance.

**Strengths:**

1. Motivation is clear.
2. The overall paper is easy to follow.

**Limitations:**

1. Could you provide more visual results compared to AR-LDM? Their visual performance and diversity are obviously better than those of CoIn, e.g., in Fig.6 and Fig.7.
2.  Could you compare the results on the StorySalon dataset with StoryGen [19]? I believe that more experiments on various datasets can support the effectiveness of your method.

I will raise my score if there are more results presented to support the CoIn.

**Suitability:**

3

---

### Official Review · Reviewer_FPNu · 2024-05-26

**Rating:** 1
**Confidence:** 3

**Summary:**

This paper proposes a lightweight and effective framework for story visualization. It introduces a context-aware story generator, and incorporates DINOv2 as the discriminator to assess the image discriminators.

**Strengths:**

- The paper presents a lightweight framework with fast processing speed.

**Limitations:**

- The writing in the paper requires significant improvement. For instance, the sentences in Lines 89-95 convey the same meaning, leading to redundancy. Additionally, similarities at the beginning of the third and fourth paragraphs indicate repetition of content.
- There are clarity issues with some figures. For example, Lines 326-328 mention Figure 3 along with the Context-aware SG and Contextual SVF, but the placement of these modules in the figure is unclear.
- The names and abbreviations of certain modules can be confusing, such as Context-aware SG, Contextual SVF, Contextual Story-Visual Fusion module, Shared Context Generator, Context G.
- The distinction between the contextual story-visual fusion and the story fusion module is unclear. The contribution of the contextual story-visual fusion module is not clearly articulated and seems to involve the use of CLIP, which may not be novel.
- The meaning of the context information in Lines 423-425 and how it is extracted using Contextual SVF are not explicitly stated.
- The choice of baselines for presenting qualitative results may not be up-to-date.
- The ablation study lacks clarity. It should involve adding each sub-module separately. The interpretation of terms like "+DINO-based D" and "Contextual SVF" is ambiguous and may lead to confusion in understanding the experimental setup. Does “+DINO-based D” mean +CLIP-based D+DINO-based D? Does “Contextual SVF” mean +CLIP-based D+DINO-based D+ Contextual G+ Contextual SVF?

**Suitability:**

3

---

### Official Review · Reviewer_fVzb · 2024-05-28

**Rating:** 5
**Confidence:** 4

**Summary:**

The paper presents CoIn, a lightweight and efficient framework for story visualization and continuation, designed to address the challenges of generating high-quality images while maintaining model efficiency. CoIn introduces three key components: a Context-aware Story Generator to capture global context information, an Intra-Story Interchange module for visual feature exchange among image generators, and DINO-based discriminators for accurate quality assessment of story images. CoIn achieves competitive image quality with significantly faster generation speeds and smaller model sizes compared to existing methods.

**Strengths:**

1. Novelty: The introduction of the Context-aware Story Generator and the Intra-Story Interchange module are quite innovative.
2. Solid Evaluation: The paper provides thorough quantitative and qualitative evaluations, including comparisons with state-of-the-art models and ablation studies that validate the effectiveness of each proposed component.
3. Promising Results: The result looks promising both in terms of generation quality and generation speed, implying the practicality of this work.

**Limitations:**

1. Would be nice to include user study as story continuation and visualization are quite subjective tasks

**Suitability:**

3

---

### Meta-Review · Area_Chair_mJ5e · 2024-06-30

**Recommendation:** Accept (Poster)
**Confidence:** 2

**Metareview:**

The paper introduces CoIn, a framework that successfully balances lightweight efficiency and high-quality image generation for story visualization and continuation.


Reasons to accept:
The paper presents effective module design for improving generation efficiency. Quantitative and qualitative evaluation support the effectiveness of the proposed approach.


Reasons to reject:
One major concern is on the writing and presentation, detailed in Reviewer FPNu’s review and response to rebuttal.
The paper is recognized for its speed advantage, although its performance is comparable to or even inferior to existing methods like AR-LDM (in WACV24, 2024/01/04). (Reviewer xhrW)
Reviewers also express concerns on the framework impact and novelty as it is limited to improve the efficiency on the specific task.


The paper receives the initial rating of weak accept, borderline reject, and reject. After rebuttal and extensive discussions, reviewers raise the rating to weak accept, borderline accept, and weak reject. After reading the paper, review comments, and discussion, AC agrees that the paper has merit and recommends accept, **with the condition that** authors incorporate the content from the rebuttal into the final version, especially the great suggestions from Reviewer FPNu on clarifying the paper presentation (in both original review and response to the rebuttal). The authors should not include the incorrect claims into the final version, such as [26] is published on 2024/04/09 while WACV24 is held on 2024/01/04.